# Selection and validation of reference genes for RT-qPCR in *ophiocordyceps sinensis* under different experimental conditions

**Li He, Jin Yi Wang, Qiang Jun Su, Zhao He Chen, Fang Xie***

School of Biological and Pharmaceutical Engineering, Lanzhou Jiaotong University, Lanzhou, GanSu, P. R. China

* xfrankf@163.com

**Data Availability Statement:** Data relevant to this study are available from Dryad at DOI:10.5061/dryad.fn2z34v1f.

## Abstract

The Chinese caterpillar mushroom, Ophiocordyceps sinensis (*O. sinensis*), is a rarely medicinal fungus in traditional chinese herbal medicine due to its unique medicinal values, and the expression stability of reference genes is essential to normalize its gene expression analysis. In this study, BestKeeper, NormFinder and geNorm, three authoritative statistical arithmetics, were applied to evaluate the expression stability of sixteen candidate reference genes (CRGs) in *O. sinensis* under different stress [low temperature (4˚C), light treatment (300 lx), NaCl (3.8%)] and different development stages (mycelia, primordia and fruit bodies) and formation of morphologic mycelium (aeriasubstrate, hyphae knot mycelium). The paired variation values indicated that two genes could be enough to accurate standardization exposed to different conditions of *O.sinensis*. Among these sixteen CRGs, 18S ribosomal RNA (*18S rRNA*) and beta-Tubulin (*β-TUB*) showed the topmost expression stability in *O.sinensis* exposed to all conditions, while glutathione hydrolase proenzym (*GGT*) and Phosphoglucose isomerase (*PGI*) showed the least expression stability. The optimal reference gene in different conditions was various. *β-TUB* and Ubiquitin (*UBQ*) were identified as the two most stable genes in different primordia developmental stage, while phosphoglucomutase (*PGM*) with elongation factor 1-alpha (*EF1-α*) and *18S rRNA* with *UBQ* were the most stably expressed for differentially morphologic mycelium stages and different stresses, respectively. These results will contribute to more accurate evaluation of the gene relative expression levels in *O.sinensis* under different conditions using the optimal reference gene in real-time quantitative PCR (RT-qPCR) analysis.

## Introduction

The Chinese cordyceps (commonly known as Dong chong xia cao) is a fungus-insect complex, which developed after Ophiocordycepss sinensis (*O.sinensis*) parasitized on the body of bat-moth larvae [1]. It is a widely known medicinal herb in the world primarily functioned as remedy for anti-cancer [2]. Pure active substances isolated from *O.sinensis*, namely polysaccharide, cordyceps acid, cordycepin and adenosine [3, 4], have been certified to possess anti-oxidant,

**Funding:** We gratefully acknowledge the financial support of the National Natural Science Foundation of China (Grant No. 31560003) and the Young Scholars Science Foundation of Lanzhou Jiaotong University (No.2022020).

**Competing interests:** The authors have declared that no competing interests exist.

anti-aging, anti-cancer and strengthen the immune system [5]. In addition, it has been certificated as a novel food source according to State Food and Drug Administration of Ministry of Public Health of China [1]. Recently, an increasing number of studies have investigated the genome of *O.sinensis* [6–8], which empowered to identify and analyze the functional genes. Characterizing these genes expression pattern will better understand the regulatory mechanisms involved in its growth and development as well as their responses to various adversity stresses.

Methods for evaluating gene expression including Northern blot, RNase protection analysis, gene chips, semi-PCR,and real-time quantitative PCR (RT-qPCR), among which RT-qPCR was generally accepted as the best effective method to quantify and validate the expression levels of target gene [9]. Nevertheless, RT-qPCR results are also often unconvinced for many interfering factors such as the integrity of RNA, the efficiency of reverse transcription, specific primers design, standardization of gene expression, and so on [10, 11]. In order to reduce the error as much as possible, one or more genes expressed stably under various conditions were selected for RT-qPCR analysis [9]. But in practice, increasing evidence has documented that numerous tested reference genes in organisms were stably expressed in specific tissues or experimental conditions [12, 13]. Furthermore, applying unstable internal reference genes will lead to significant differences in RT-qPCR results [14]. Therefore, normalization reference gene is essential for ensuring the reliability of RT-qPCR results in different tissues or conditions.

A growing number of evidence has been shown on the screening of reference genes in macro-fungi, such as Ganoderma lucidum, *Auricularia cornea*, *Lentinula edodes*, *Pleurotus ostreatus*. Some housekeeping genes, like actin rehulatory protein 1 (*ACT1*), phosphoglucose mutase (*PGM*), β-tubulin (*β-tub*), 18S ribosomal RNA (*18s rRNA*) were proved to be the most stable internal reference genes in these organisms [15–18]. For Cordycepss species (T.guangdongense), there was only one study reported that *vps* and histone (*H4*) have been identified to be as the preferred reference genes, whereas tubulin beta chain 2 (*β-tub2*) was the least stable gene across all conditions, and 40S-18s (*40S*) was considered to be the most stable gene when exposed to different carbon source conditions [19]. But so far no such investigations available for *O.sinensis* related to the reference genes used for RT-qPCR normalization.

It is universally acknowledged that the initiation and formation of stroma is a key factor in the growth of *O.sinensis*, and the study on mechanism and change process during this period is the basis for artificial cultivation in Chinese cordyceps. In addition, our team observed that three distinct mycelial morphologies arose during the process of *O.sinensis* cultivation [20]. Meanwhile, in our recently studies, urea cycle, purine metabolic and carotenoid metabolic pathway were found to be a close response to light treatment [21], and we also work on its response mechanism to low temperature, drought and NaCl treatment (unpublished data). Nevertheless, the in-depth understanding of molecule mechanism for the response of *O.sinensis* to various adverse situations as well as materials of developmental stages was scarce. So it was crucial to identify the optimal reference genes for gene expression analyses across all situations.

In this study, sixteen candidate reference genes (CRGs) were screened from *O.sinensis* transcriptome data (unpublished data). The stability expressions of these CRGs were analyzed by geNorm, NormFinder and BestKeeper under different stress (4°C low temperature, light treatment, 3.8% NaCl) and different development stages and formation of morphologic mycelium. This research was designed to screen out the most preferred reference genes for a wide range of gene expression analysis in the presence of all situations, which will provide important reference basis for omics research of *O.sinensis*.

## Materials and methods

### Strain, culture conditions and sample collection

The *O.sinensis* strain tz8-1 applied in this study was cultivated according to the method of single ascospore [21]. For abiotic stress treatment group, mycelium of *O.sinensis* strain tz 8–1 cultivated on milk medium under 18°C for 45 d and then divided into seven groups, the first and second group were exposed to light (3000lx) and dark (control) for 10d, respectively. The third group treated with 3.8% NaCl and without treatment (CK) for 20 d, respectively. The fifth and sixth group were treated under 4°C (low temperature stress) and 18°C (CK). 80 mg different treatment materials were taken and put them in liquid nitrogen, then placed in -80°C refrigerator, immediately. Three biological duplicate samples from each treatment group. In addition, three morphologic mycelium, primordia and fruiting bodies of tz 8–1 were also collected as the method of Xie's study [21].

### RNA isolation and cDNA synthesis

RNA extraction was carried out according to the method described by Xie et al [7, 21]. TRIzol reagent (mir VanaTMmiRNA ISOlation Kit, Ambion-1561) was applied to extract total RNA from samples, RNA concentration and OD260/OD280 were detected using the NanoDrop 2000 spectrophotometer. RNA integrity was detected with agarose gel electrophoresis, then using TransScript All-in-One First-Strand cDNA Synthesis SuperMIX for qPCR Kit for cDNA synthesis. The reverse transcription system contained 0.5 μL RNA, 5μL 5×TransScript All-in-One SuperMIX for qPCR; 0.5 μL gDNA Remover and 4 μL nuclease-free $H_2O$. Ampliffcation conditions were as follows: 42°C 15min, 85°C 5s, then added 90 mL nucleasefree $H_2O$ and stored in refrigerator at -20°C for further use.

### Real time PCR analysis

The RT-qPCR reaction was performed on lightcycle® 480 II fluorescence quantitative PCR (Roche, Swiss) using PerfectStartTM Green qPCR SuperMix Kit. PCR reaction system were as follows: 5μL 2×PerfectStartTM Green qPCR SuperMix; 0.2 μL,10μM Forward primer; 0.2 μL,10μM Reverse primer; 1 μL cDNA and 3.6 μL nucleasefree $H_2O$. Ampliffcation conditions were as follows: 94°C 30s; 94°C 5s; 60°C 30s; 45 cycles, then extension at 72°C for 5 min, and finally detected the product specificity by melting curve. Choosing *18S rRNA* as the internal reference gene to analyze results with the calculation method of $2^{-\Delta\Delta CT}$ [7].

### Selection of candidate reference genes and primer design

Based on our recently transcriptome analysis data, sixteen genes sequences downloaded from the genome database of *O.sinensis* (ASM44836v1) [21] were used for primer design. More detailed information of these genes was available in Table 1.

### Expression stability analysis

After RT-qPCR amplification completed, using geNorm [22], NormFinder [23] and Best-Keeper [24] software to analyze the expression differences of sixteen candidate reference genes (CRGs). Finally using RefFinder [25] software comprehensively ranked the expression stability of sixteen CRGs.

**Table 1. Primers were designed with PrimerPremier5.0.**

| Symbol | Gene ID | Forward primer sequence | Reverse primer sequence | Size | Efficiency | $R^2$ |
|---|---|---|---|---|---|---|
| 18S RRNA | CCM850883 | GCAGTGGCATCTCTCAGTC | TCATCGATGCCAGAACC | 128 | 96.40% | 0.993 |
| QPRTase | CCM23475 | ATGCTGAGCTGTTTAGCG | TGCCCTGTTCCGTCGTAGA | 90 | 102.23% | 0.988 |
| β-TUB | CCM37238 | TACGCCTCTTCGACGATAG | GCCGTTGTACACCATT | 148 | 93.67% | 0.995 |
| RPL2 | CCM845215 | ACCTACCGTCTCCATCAT | GTGAACCTGCTGGACAAT | 109 | 99.09% | 0.989 |
| EF1-a | CCM36271 | CAAGGGCTCTTTCAAGTATGC | GTGACATAGTACCTGGGAGT | 114 | 92.34% | 0.997 |
| PGI | CCM 35886 | TTCGACCAGTATCTTCATCGC | GTGTACTTGACCGACGATCC | 83 | 104.37% | 0.994 |
| PGM | CCM835257 | AAGCCCTTTCAGGACCAA | GAACGACTCGGTGTAGTG | 87 | 96.64% | 0.995 |
| H$^+$-ATPase | CCM547525 | CGCTTCGCGGAAATCTATAC | AGACGTTCTTGTTGACGG | 90 | 98.79% | 0.993 |
| ACT1 | CCM818339 | CAATCGGCACAACTGGACA | GACGACCTGAGCGGAATA | 95 | 100.23% | 0.996 |
| UBQ | CCM175840 | CGACATCGAGTTGGACTAC | ATACCTGCAATCTGTCCG | 82 | 105.34% | 0.984 |
| GAPDH | CCM2597 | GAGGCCGAGAGCCAACTA | TTCATCACGACAGCACCA | 99 | 97.77% | 0.988 |
| CYS | CCM09645 | TGGCCTTCGCTCTAAAGTG | TCGCCTATACGAGACCCA | 123 | 95.54% | 0.987 |
| GGT | CCM947947 | TGAGCAACTCGTTCGGCTA | GACGATGATGGGCGAGAT | 134 | 98.97% | 0.989 |
| TPI | CCM43582 | TAACAGTCTTGTCTCGACCG | GCCTTGACAAACAATGGCA | 107 | 102.29% | 0.995 |
| TYR1 | CCM641512 | TGCGTGGTGGAGCATGAA | ATCCACAACGGAATACACTG | 115 | 103.34% | 0.988 |
| CdC14 | CCM34067 | ATATTACCCGTTGGTTGTGG | CAAAGGAGGAAGGCGTGA | 143 | 100.45% | 0.994 |

## Results

### Selection of candidate reference genes and primer design

Sixteen CRGs were those eleven house-keeping genes involving 18S ribosomal RNA (*18S rRNA*), adenine phosphoribose transferase (*QPRTase*), microtubules protein (*β-TUB*), Ribosomal protein L2 (*RPL2*), Translation elongation factor 1-α (*EF1-α*), Phosphoglucose Isomerase (PGI), Phosphoglucose mutase (*PGM*), Plasma membrane proton ATPase (*H$^+$-ATPase*), actin 1(*ACT1*), Polyubiquitin enzyme (*UBQ*), Glycerol 3-phosphate dehydrogenase (*GAPDH*), and five candidate genes including transcription factor (*CYS*), glutathione hydrolase proenzyme (*GGT*), triose phosphate isomerase (*TPI*), pro-phenol oxidase subunit 1 (*TYR1*) and phosphoprotein phosphatase (*CDC14*). The detailed information of these CRGs was shown in Table 1. Furthermore, BLAST analysis indicated that the primers of these CRGs were specific matched to the target genes sequence in the genome of *O.sinensis*. The results of dissolution curve analysis indicated a unique peak for each PCR product (S1A Fig). Furthermore, agarose gel electrophoresis results showed all of sixteen CRGs were specifcally amplifed a single product with specific size (S1B Fig), exhibiting the excellent specificity for all primers designed in this study.

### Expression profile of sixteen candidate reference genes in *O.sinensis*

The $C_t$ values of sixteen CRGs were standardized for RT-qPCR to produce a standard curve with 10-fold serial cDNA diluents. The PCR efficiencies (E) were counted with theorem $E = 10^{-1/slope} - 1$ varied from 92.34% for *EF1-α* to 104.37% for *PGI* (Table 1). The correlation coefficients ($R^2$) of these sixteen CRGs were higher than 0.98 (Table 1), suggesting the each $C_t$ values had a credible linear relation. The average of $C_t$ values of all these CRGs in all experimental conditions varied from 15.43 for *CDC14* to 30.12 for *UBQ* (Fig 1A), among which average $C_t$ values of 11 CRGs were lower than 25, involving *β-TUB*, *RPL2*, *EF1-α*, *PGI*, *PGM*, *18S rRNA*, *CYS*, *GGT*, *TPI*, *TYR1* and *CDC14*, indicating these genes were highly expressed in all samples. *CDC14* was the most highly expressed CRGs with low $C_t$ values, while *UBQ* was the least expressed CRGs with highest $C_t$ values in all samples.

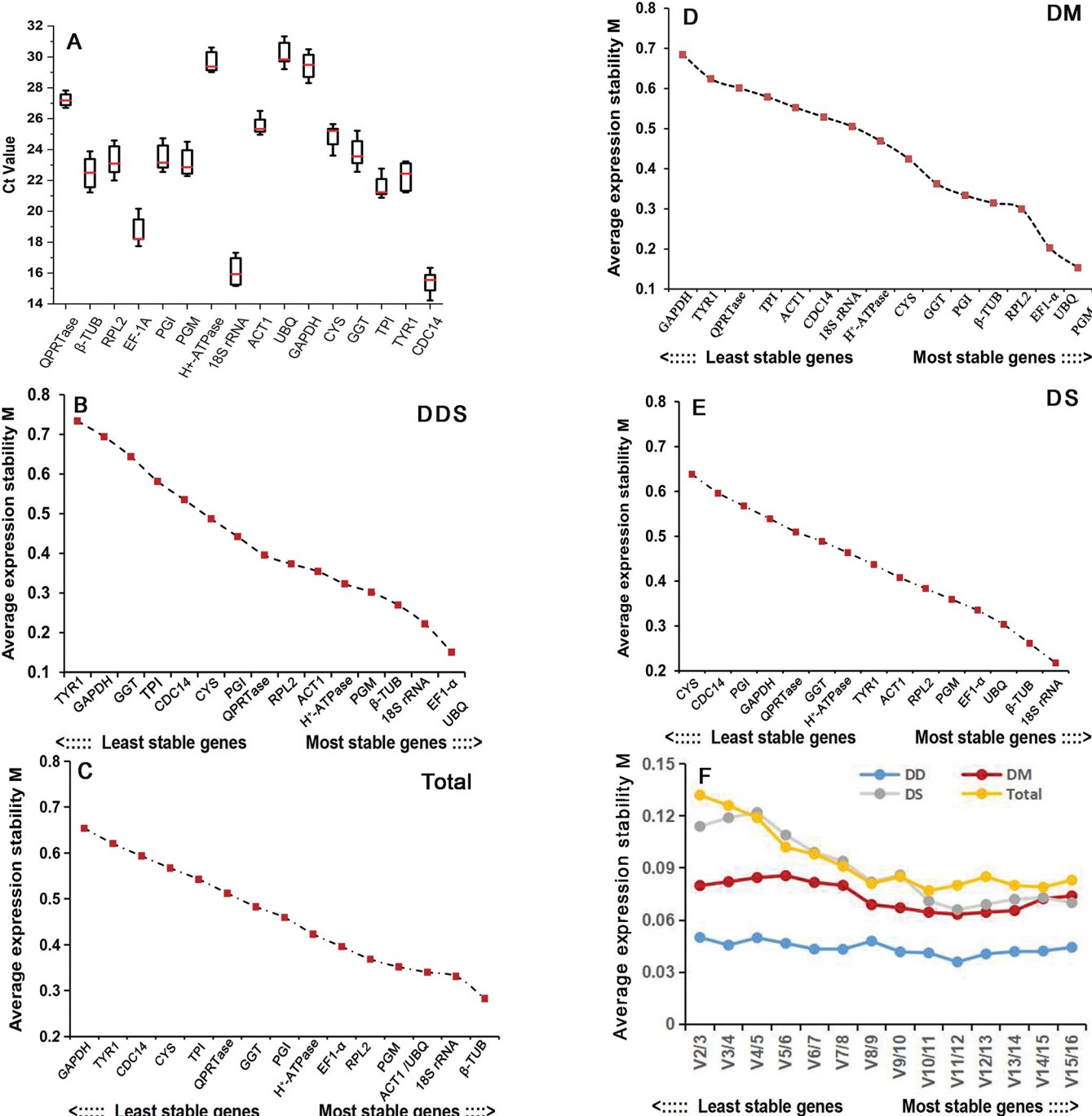

**Fig 1. Expression stability analysis of sixteen candidate reference genes in *O. sinensis*.** (A) variation in RT-qPCR values of nineteen candidate reference genes (CRGs) in all samples, (B-E) expression stability ranking of sixteen CRGs using geNorm under different development stages (mycelia, primordia, young and mature fruiting bodies), total samples, different temperature stresses (heat and cold) and all samples, respectively. (F) pairwise variation (Vn/Vn+1) exhibiting the optional number of reference genes required in an accurate normalization. DDS, differentially developmental stages; Total,total samples;DM, Differentially Morphologic Mycelium Stages; DS, Different Stresses.

## GeNorm analysis of sixteen CRGs in *O.sinensis*

Based on RT-qPCR $C_t$ values, GeNorm statistical algorithms was introduced to assess the stability of CRGs by counting the normalization factor. The M value was described as the average variation of some genes from all other potential CRGs. The genes with smaller M values will show higher expression stability profile. As shown in Fig 1B, during differentially developmental stages of primordia, *EF1-α* and *UBQ* with the smallest value of 0.157 were identified as the most steadiest genes, followed by *β-TUB* and *18S rRNA*, compared to the most unstable genes of *TYR1* and *GAPDH* with M values of 0.732 and 0.692, respectively. Under differentially morphologic mycelium stages, *UBQ* and *PGM* with the smallest M value of 0.162 were the most steadily expressed, whereas *GAPDH* and *TYR1* were the least stable (Fig 1D). When exposed to different stresses, *β-TUB* and *18S rRNA* with M value of 0.215 were the most steadiest genes, while *CYS* and *CDC14* were the most unstably expressed (Fig 1E). But the total data demonstrated that *β-TUB* and *18S rRNA* were the most stably expressed genes, followed by *ACT1* and *UBQ*, whereas *GAPDH* and *TYR1* were the most unstably expressed (Fig 1C). The paired variation values ($V_n/V_{n+1}$) of sixteen CRGs were calculated with the geNorm algorithm to identify the number of options available reference genes for accurate norming. According to previous research reports [22], the extra reference gene would be desired when its pairwise variation value was less than 0.15 (Fig 1F and S1 Table), indicating that two genes stably expressed were adequate for precise normalization for *O. sinensis* under various conditions. The optimal combination of reference genes was *EF1-α* and *UBQ* in different development stages, *UBQ* and *PGM* for differentially mycelial developmental stages, *β-TUB* and *18S rRNA* for different stresses, respectively. Whereas, *β-TUB* and *18S rRNA* were the optimal combination of reference genes under conditions (S1 Table).

## NormFinder analysis of sixteen CRGs in *O.sinensis*

In order to explore suitable candidate genes for normalization, the NormFinder method was used to rank CRGs [26]. NormFinder analysis results showed that the most stable gene was *ACT1* for different primordia developmental stages, *PGM* for differentially morphologic mycelium stages, *UBQ* for different stresses, while 18S rRNA for total conditions. Meanwhile, ranked in the second and third positions were *β-TUB* and *18S rRNA* for different primordia developmental stages, *EF1-α* and *UBQ* for differentially morphologic mycelium stages, *18S rRNA* and *EF1-α* for different stresses, *β-TUB* and *PGM* for total conditions. The least stable genes were *CYS* and *CDC14* for differentially morphologic mycelium stages and total conditions, while *GAPDH* for different primordia developmental stages and stresses (Table 2).

## BestKeeper analysis of sixteen CRGs in *O.sinensis*

BestKeeper method was used to determine the most stable housekeeping genes by using repeated pair-wise correlation analysis based on the sum of stability values in geNorm and NormFinder analyses, the top 10 genes were chosen for further analysis [24]. The standard deviation (SD) and correlation coefficient (R) values of sixteen CRGs were introduced to evaluate its expression stability. The smaller the values of SD and CV, the more stable of CRGs, and which was unstable when SD value was larger than one [27]. As shown in Table 3, *β-TUB* and *UBQ* were identified as the most stable genes for different primordia developmental stages, while *GGT* and *PGI* with a higher SD value (>1) and lower R value were identified as the least stable genes. In terms of differentially morphologic mycelium stages, the most stably expressed genes were *PGM*, followed by *EF1-α* and *UBQ*, whereas *GGT* and *TPI* were the unstably genes with higher SD value. In addition, *18S rRNA* was the most stably expressed genes when exposed to different stress, followed by *UBQ* and *EF1-α* as the second and third positions, respectively. Nevertheless, $H^+$-*ATPase* was eliminated due to its higher SD value

**Table 2. Stability analysis of sixteen CRGses by NormFinder.**

| Ranking | Different Primordia Developmental Stages | | Differentially Morphologic Mycelium Stages | | Different Stresses | | Total | |
|---|---|---|---|---|---|---|---|---|
| | NormFinder | Stability Value | NormFinder | Stability Value | NormFinder | Stability Value | NormFinder | Stability Value |
| 1 | ACT1 | 0.093 | PGM | 0.113 | UBQ | 0.101 | 18S rRNA | 0.261 |
| 2 | β-TUB | 0.153 | EF1-α | 0.132 | 18S rRNA | 0.125 | β-TUB | 0.261 |
| 3 | 18S rRNA | 0.179 | UBQ | 0.153 | EF1-α | 0.141 | PGM | 0.315 |
| 4 | RPL2 | 0.242 | β-TUB | 0.204 | β-TUB | 0.169 | UBQ | 0.349 |
| 5 | UBQ | 0.266 | PGI | 0.233 | ACT1 | 0.213 | EF1-α | 0.382 |
| 6 | EF1-α | 0.284 | RPL2 | 0.269 | RPL2 | 0.247 | ACT1 | 0.412 |
| 7 | PGM | 0.344 | H+-ATPase | 0.284 | PGM | 0.272 | H+-ATPase | 0.447 |
| 8 | TPI | 0.359 | 18S rRNA | 0.305 | H+-ATPase | 0.279 | RPL2 | 0.462 |
| 9 | QPRTase | 0.387 | GGT | 0.362 | CYS | 0.318 | TPI | 0.482 |
| 10 | H+-ATPase | 0.442 | ACT1 | 0.404 | CDC14 | 0.344 | QPRTase | 0.509 |
| 11 | PGI | 0.467 | TPI | 0.479 | TPI | 0.459 | PGI | 0.533 |
| 12 | GGT | 0.479 | QPRTase | 0.501 | QPRTase | 0.497 | CYS | 0.558 |
| 13 | TYR1 | 0.485 | TYR1 | 0.524 | PGI | 0.592 | GGT | 0.586 |
| 14 | CDC14 | 0.504 | GAPDH | 0.554 | TYR1 | 0.611 | GAPDH | 0.610 |
| 15 | CYS | 0.517 | CDC14 | 0.566 | GGT | 0.639 | TYR1 | 0.633 |
| 16 | GAPDH | 0.539 | CYS | 0.572 | GAPDH | 0.661 | CDC14 | 0.658 |

**Table 3. Stability analysis of sixteen CRGs calculated by the BestKeeper.**

| DDS | | | | | | | | | | |
|---|---|---|---|---|---|---|---|---|---|---|
| | ACT1 | β-TUB | 18S rRNA | RPL2 | UBQ | EF1-α | TPI | QPRTase | PGI | GGT |
| std dev[±CP] | 0.66 | 0.33 | 0.60 | 0.76 | 0.41 | 0.57 | 0.84 | 0.89 | 0.92 | 1.12 |
| CV[%CP] | 1.56 | 2.14 | 1.89 | 2.69 | 3.01 | 2.19 | 3.19 | 1.43 | 1.55 | 2.77 |
| Coeff.of corr.[r] | 0.79 | 0.95 | 0.88 | 0.81 | 0.93 | 0.92 | 0.82 | 0.71 | 0.60 | 0.79 |
| Ranking | 5 | 1 | 4 | 6 | 2 | 3 | 7 | 8 | 9 | / |
| DM | | | | | | | | | | |
| | β-TUB | 18S rRNA | RPL2 | PGM | EF1-α | TPI | PGI | GGT | UBQ | H+-ATPase |
| std dev[±CP] | 0.48 | 0.53 | 0.77 | 0.39 | 0.33 | 0.96 | 0.63 | 0.89 | 0.26 | 0.81 |
| CV[%CP] | 2.11 | 3.25 | 1.98 | 2.44 | 3.44 | 2.66 | 1.66 | 2.88 | 1.54 | 3.38 |
| Coeff.of corr.[r] | 0.90 | 0.82 | 0.80 | 0.93 | 0.92 | 0.77 | 0.72 | 0.71 | 0.94 | 0.71 |
| Ranking | 4 | 5 | 7 | 2 | 3 | 10 | 6 | 9 | 1 | 8 |
| DS | | | | | | | | | | |
| | β-TUB | 18S rRNA | RPL2 | UBQ | EF1-α | PGI | GGT | PGM | H+-ATPase | ACT1 |
| std dev[±CP] | 0.45 | 0.29 | 0.77 | 0.34 | 0.41 | 0.93 | 0.85 | 0.68 | 1.17 | 0.56 |
| CV[%CP] | 2.19 | 1.67 | 3.56 | 4.11 | 1.67 | 2.56 | 3.11 | 2.51 | 1.88 | 2.77 |
| Coeff.of corr.[r] | 0.91 | 0.97 | 0.85 | 0.81 | 0.87 | 0.77 | 0.83 | 0.72 | 0.63 | 0.67 |
| Ranking | 4 | 1 | 7 | 2 | 3 | 9 | 8 | 6 | / | 5 |
| TT | | | | | | | | | | |
| | β-TUB | 18S rRNA | RPL2 | UBQ | EF1-α | PGI | GGT | PGM | H+-ATPase | ACT1 |
| std dev[±CP] | 0.44 | 0.36 | 0.77 | 0.49 | 0.50 | 0.88 | 0.96 | 0.66 | 0.81 | 0.74 |
| CV[%CP] | 3.11 | 2.67 | 1.87 | 2.33 | 1.77 | 3.99 | 4.22 | 3.09 | 2.19 | 2.56 |
| Coeff.of corr.[r] | 0.90 | 0.97 | 0.93 | 0.86 | 0.92 | 0.82 | 0.77 | 0.72 | 0.71 | 0.75 |
| Ranking | 2 | 1 | 8 | 3 | 4 | 9 | 10 | 5 | 8 | 6 |

/ indicated that gene unranked based on the BestKeeper analysis. DDS, Different Primordia Developmental Stages; DM, Differentially Morphologic Mycelium Stages; DS, Different Stresses; TT, Total samples.

(1.12) and low R value (0.67). From the perspectives of total samples, *18S rRNA* and *β-TUB* owing higher SD value and low R value were taken as the stably expressed genes.

Based on the above three statistical results, *β-TUB* and *UBQ* were identified as the two most stable genes in different primordia developmental stage, while *PGM* with *EF1-α* and *18S rRNA* with *UBQ* were the most stably expressed for differentially morphologic mycelium stages and stresses, respectively. Collectively, *18S rRNA* and *β-TUB* could be selected as the most optimal reference gene under different conditions.

## Validation of selected reference genes in *O.sinensis*

To verify the reliability of these results, the relative expression level of nine target genes were evaluated using three reference genes groups, including the most stable, combination and the least stable genes under different experimental conditions. It was observed that the relative expression levels of three tested genes showed significant differences normalized with different reference genes exposed to different developmental stages (Fig 2A). Compared to mycelium stage, cytochrome oxidase 2 (*Oscox2*) normalized by *β-TUB* was obviously upregulated under primordium and fruiting body stage, while there was nearly no significant difference observed in the analysis with *UBQ* and their combination as the reference genes, let alone *GGT*. Meanwhile, we found that Alt-like RNA polymerase ADP-ribosyltransferase 4 (*Osnad4*) and ATP synthase, beta subunit 8 (*Osatp8*) normalized by *β-TUB*, *UBQ* and their combination were markedly upregulated in primordium stage, then exhibited a decreasing trend in fruiting body stage, but no significant differences were detected with the reference gene *GGT* at any stage. The above results confirm that *β-TUB* and *UBQ* were the most stable reference genes at different development stages.From the perspectives of differentially morphologic mycelium stages three genes displayed striking differences of the expression levels with different genes as reference genes (Fig 2B). The relative expression level of ribosomal protein S3 (*Osrps3*) normalized by *PGM* exhibited asignificant upregulation in hyphae knot stage liken to aerial mycelium, then the expression was significantly down-regulated at substrate mycelium, it also showed similar trend normalized by *EF1-α* or the combination of *PGM* and *EF1-α*. When normalized with the less stable gene *TPI*, it was expressed at lower levels and no significantly variation. Furthermore, the expression pattern of ribosomal protein S115 (*Osrps115*) was similar to *Osrps3* except for higher expression, whereas no difference was observed in expression level of *Oscox1* normalized with different reference gene at different mycelium stages. Thus, we inferred that *PGM* and *EF1-α* were the most stable reference genes at different mycelium stage.

Under low temperature stress, an obvious difference was observed in RT-qPCR normalized by different reference genes (Fig 2C). It was found that the relative expression levels of *Oscox3* and actinodin 2 (*Osand2*) normalized by *18S rRNA*, *UBQ* and its combination were all markedly upregulated when exposed to cold stress (4˚C) compared with normal temperature (18˚C), and then noticeably downregulated in the presence of extremely cold stress (-2˚C), while no difference was seen normalized using the less stable gene $H^+$-*ATPase* under different temperature conditions. In addition, the relative expression levels of cytochrome b (*Oscob*) showed a similar variation trend with the two former reference genes using different reference genes, but just no significant difference. Consequently, the selection of inappropriate reference genes could lead to incorrect assessment of relative transcript abundance, which can lead to the biased results. It was inferred that *18S rRNA* and *UBQ* were the most stable reference genes at different stress.

## Discussion

In recent years, although siious stresses was imperative to address these problems. RT-qPCR was identified as the best effective method to understand the molecular mechanism by

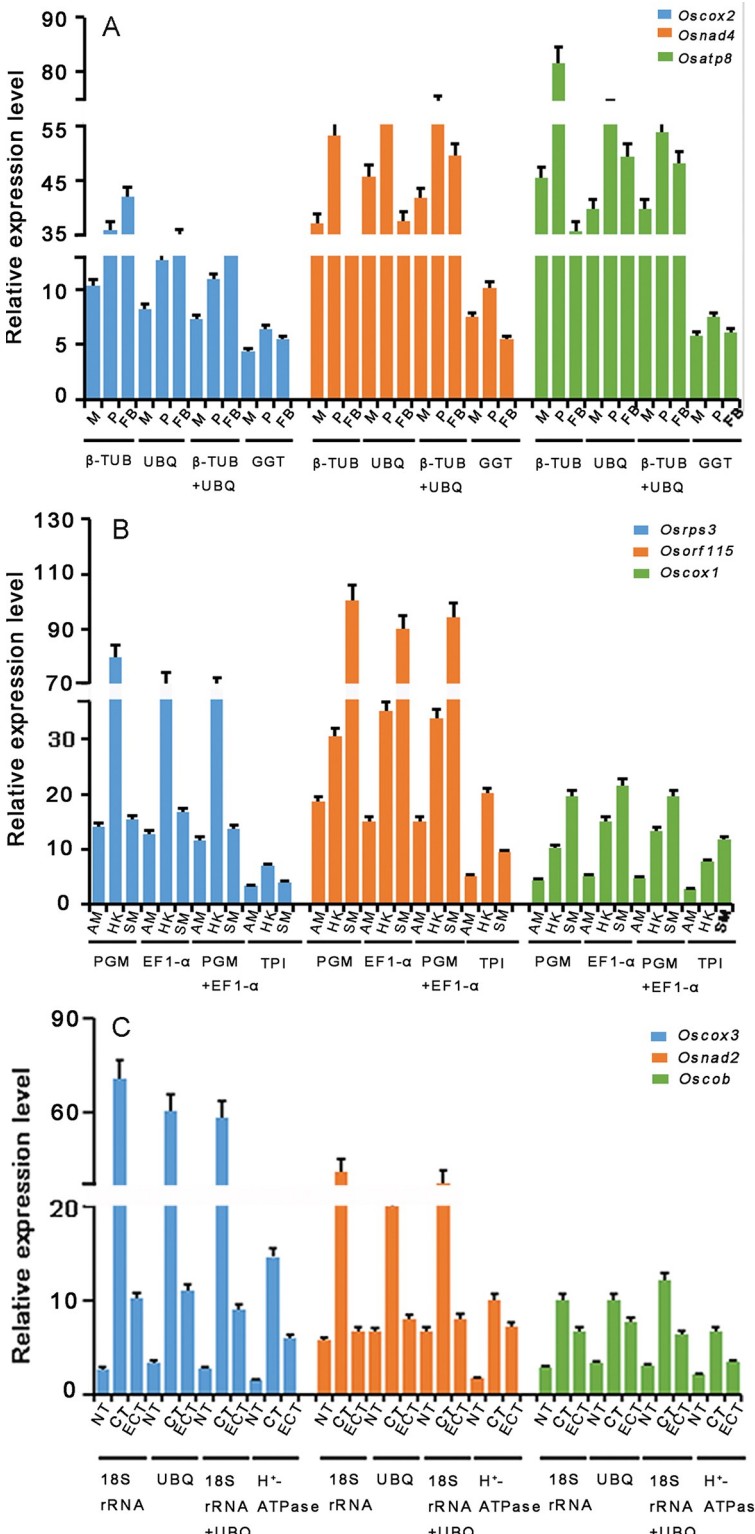

**Fig 2. The relative expression levels of nine genes normalized by the selected reference genes under different testing conditions.** (A) RT-qPCR analysis of three genes normalized using the most stable reference genes (*β-TUB* or *UBQ*) or a combination (*β-TUB* and *UBQ*) and the least stable genes (*GGT*) during different development stages. (B) RT-qPCR analysis of three genes normalized using the most stable reference genes (*PGM* or *EF1-α*) or a combination (*PGM* and *EF1-α*) and the least stable genes (*TPI*) under differentially morphologic mycelium stages. (C) RT-qPCR

analysis of three genes normalized using the most stable reference genes (*18S rRNA or UBQ*) or a combination (*18S rRNA* and *UBQ*) and the least stable genes (*H⁺-ATPase*) under cold stress (-2°C). Error bars indicate the mean standard error calculated from three biological replicates. The statistical level was according to * P < 0.05, **P < 0.01.

quantifying the expression of key genes [9, 19]. However, stable reference genes were a precondition for measuring the accuracy of RT-qPCR results [28]. Numerous published studies have been dedicated to the identification of reference genes in organisms [16, 29–35], among which some house-keeping genes were certified to be the unstable expression under various experimental conditions [16, 33–35]. So it was urgently necessary to recruit the optimal reference genes for RT-qPCR analysis. From the point of macro-fungi, the conventional and original candidate gene has been evaluated in *Auricularia cornea*, *Cordyceps militaris*, *Ganoderma lucidum* under different developmental stages, medium and abiotic stresses [16, 19, 36, 37]. These reports indicated that the selection of the best reference genes should be based on specific experimental conditions, sample materials and even strains.

In this study, an integrated analysis of sixteen CRGs, including conventional and original candidate genes from previous research, was carried out under different primordia development stages (mycelium, primordia and fruit bodies), differentially morphologic mycelium stages (aerial mycelium, hyphae knot and substrate mycelium), different stresses (4°C temperature, 3.8% NaCl and light treatment). The melting curves as well as $R^2$ and E values were all demonstrated high specificity and amplification efficiency for these sixteen CRGs (Fig 1 and Table 1).

Under different primordia development stages, *β-TUB* was identified as the most stable gene based on three statistical analyses, which was consistent with previous studies in fungi (*T. guangdongense* and *L.edodes*) and other organisms [34, 35, 38–40]. *UBQ* encoding small subunit ribosomal protein, was demonstrated as second best reference gene by three statistical analyses on the basis of its biological function, and it had been proven to be the most stably expressed in Cordyceps guangdongensis and *Gentiana macrophylla* when exposed to different temperature treatments [41, 42], suggesting *UBQ* as the candidate reference gene could be reliable in different sample materials. With regards to differentially morphologic mycelium stages, *UBQ* was also confirmed to be the most stably expressed gene, and *PGM* encoding phosphoglucomutase ranked in second position according to the three algorithm programs, it has rarely been reported as an optimal reference gene in previous research.

For different stress conditions, the selection of reference genes has been studied in various edible fungus. Such as in *L.edodes*, *N.tabacum* and *C.capsularis*, *actin* was confirmed as the best stably expressed gene in subjected to various abiotic stresses [43–45], while cytochrome c heme lyase (*CYC3*) and vesicle-fusing ATPase (*VPS4*) were identified as reliable reference gene for Morchella sp. T. guangdongense under temperature stress [41]. However, in our study, *18S rRNA* was verified to be the most suitable reference genes, which was consistent with the previous study that it was observed to be the most stable expression gene in all samples [15, 17, 46]. Meanwhile, *UBQ* was also defined as the second-stable expressed genes in different stress conditions. Due to the pairwise variation $V_{2/3}$ values were all below 0.15, indicating these two genes were appropriate to normalize gene expression analysis. Thus, *18S rRNA* with *UBQ* was recommended as the pair of reference genes for *O.sinensis* in the presence of various stresses.

It has been reported that two reference genes were required to normalize gene expression in RT-qPCR analysis when the paired variation values are both less than 0.15 [26]. Based on the overall analysis results, *β-TUB*, *18S rRNA*, *UBQ* and *EF1-α* were found to be the top four candidate reference genes of sixteen genes. Taking into account all the experimental

conditions, *18S rRNA* and *β-TUB* were ranked in the first two positions based on the analyses of the three algorithm programs, suggesting that these two genes were credible for standardization of gene expression analysis in this study. These results will better explore the molecular mechanisms on this valuable medicinal herb.

## Supporting information

**S1 Fig. Dissolution curves and amplification products and of sixteen CRGs.** A. Dissolution curves and amplification products and of sixteen CRGs. B. Amplifcation products of 16 CRGs: DNA marker, DL2000; 1–16, *18S rRNA*, *QPRTase*, *β-TUB*, *RPL2*, *EF1-α*, PGI, *PGM*, $H^+$-ATPase, *ACT1*, *UBQ*, *GAPDH*, *CYS*, *GGT*, *TPI*, *TYR1*and *CDC14*.
(TIF)

**S1 Table. The optimal combination of reference genes under different conditions.**
(PDF)

## Acknowledgments

We would like to thank the Biomedical Technology Co from Shanghai Ouyi for special fungal technical support and remote guidance.

## Author Contributions

**Conceptualization:** Li He, Fang Xie.

**Data curation:** Li He.

**Funding acquisition:** Fang Xie.

**Methodology:** Fang Xie.

**Resources:** Qiang Jun Su, Zhao He Chen.

**Software:** Jin Yi Wang.

**Supervision:** Fang Xie.

**Validation:** Li He, Jin Yi Wang, Qiang Jun Su, Zhao He Chen, Fang Xie.

**Visualization:** Li He, Jin Yi Wang, Qiang Jun Su, Zhao He Chen, Fang Xie.

**Writing – original draft:** Li He, Jin Yi Wang, Zhao He Chen.

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
