## [Decision Letter · Decision Letter 0]

17 May 2023

PONE-D-22-28076Selection and Validation of Reference Genes for qRT-PCR in Ophiocordyceps sinensis under Diﬀerent Experimental ConditionsPLOS ONE

Dear Dr. Li,

Thank you for submitting your manuscript to PLOS ONE. After careful consideration, we feel that it has merit but does not fully meet PLOS ONE’s publication criteria as it currently stands. Therefore, we invite you to submit a revised version of the manuscript that addresses the points raised during the review process.

Please address review comments, explain in detail if reviewers' suggestion/s are not followed.

We look forward to receiving your revised manuscript.

Kind regards,

Xiaoping Pan

Academic Editor

PLOS ONE

https://www.mdpi.com/2073-4425/10/9/647/htm

https://www.sciencedirect.com/science/article/abs/pii/S0378111920300494?via%3Dihub

In your revision ensure you cite all your sources (including your own works), and quote or rephrase any duplicated text outside the methods section. Further consideration is dependent on these concerns being addressed.

“We gratefully acknowledge the ﬁnancial support of the National Natural Science Foundation

of China (Grant No. 31560003). We are so grateful for language modiﬁcation of Yuexia Wang, Ph.D.”

 “The author(s) received no specific funding for this work”

Reviewers' comments:

Reviewer's Responses to Questions

**Comments to the Author**

1. Is the manuscript technically sound, and do the data support the conclusions?

Reviewer #1: Yes

Reviewer #2: Partly

2. Has the statistical analysis been performed appropriately and rigorously? 

Reviewer #1: No

Reviewer #2: Yes

3. Have the authors made all data underlying the findings in their manuscript fully available?

Reviewer #1: Yes

Reviewer #2: No

4. Is the manuscript presented in an intelligible fashion and written in standard English?

Reviewer #1: Yes

Reviewer #2: No

5. Review Comments to the Author

Reviewer #1: The authors reported the identification of reference genes for RT-qPCR analysis in the fungus Ophiocordyceps sinensis. The work has been carried out in a correct and “classical” way for this type of manuscript and reports interesting and useful results for those who deal with this fungus.

I suggest some improvements:

In M&M, please indicate clearly how many biological replicates have been analysed, and how the biological replicates were obtained.

Fig 3: add statistical analysis

Throughout the manuscript, please change qRT-PCR to RT-qPCR.

In the abstract, indicate in full the names of the genes.

Reviewer #2: The manuscript “Selection and Validation of Reference Genes for qRT-PCR in Ophiocordyceps sinensis under Different Experimental Conditions” is a typical reference gene finding study on the medicinal fungus. Although the study is essential for future molecular work, the way the results are present ed is not optimal. I am pointing out some of the major issues with the manuscript that need to be resolved for consideration for publication in PLOS One.

Major issues:

1. The expression of 16 candidate genes must be supplied from the transcriptome data in the present study. The sequence of 16 selected transcripts needs to be given as supplementary.

2. How much amount of RNA was taken for cDNA synthesis?

3. I assume there must be some published references for RNA extraction in this specie, as the genome is already available. Can you cite them in the material method section?

4. 2-delta delta Ct reference missing.

5. I recommend adding a table with 16 selected genes and their transcriptomic expression in the different conditions in Ophiocordyceps sinensis (as per availability) with a putative function and reference for the readers.

6. Line 123- 130 is unnecessary as the full name of the genes is given in the table1.

7. The melting curve picture should be supplementary. Try to give an agarose gel picture of the PCR-amplified target genes (16) as a supplementary.

8. Line 243: either write other fungal species or if you want to make a general comment like “in organisms” then add some recent citations: https://doi.org/10.3389/fphys.2021.752768 ; https://www.mdpi.com/1422-0067/24/1/451 ; https://www.nature.com/articles/s41598-021-92780-1 ; https://arxiv.org/abs/2301.04653;
https://doi.org/10.1016/j.jplph.2023.153925 ;etc

9. There is some redundancy in different manuscript sections; try to minimize those. For example, line 277 and line 281 redundancy mentions the same thing.

10. Mention the function of the Oscox gene somewhere in the manuscript to prepare the ground for the readers to digest the result and the rationality of it during the validation of the selected gene section.

Minor issues:

1. English is very poor. I recommend professional linguistic correction before resubmission.

2. The short forms of gene names for less common genes are often hard to follow for non-field readers.

3. The species names should be in Italics throughout the text.

4. Do not use the short form in the abstract. Alternatively, short forms must appear with the full form in the first appearance in the text.

5. Line 43: qPCR or qRT-PCR?

6. Line 47: PCR or qRt-PCR?

7. Line 91: Xie`s study? Please refer to the article.

8. Word spacing needs to be corrected throughout the text. The authors are not very carefully edit the manuscript before submission. (i.e., line 26, 89,199)

9. Line 198: correct (“and and”)

10. Line 206: validation section: try to mention the most stable or least stable gene in the bracket while explaining the result. It is interesting but hard to follow in its present form.

6. PLOS authors have the option to publish the peer review history of their article (what does this mean?). If published, this will include your full peer review and any attached files.

Reviewer #1: No

Reviewer #2: No

---

## [Author Response · Author response to Decision Letter 0]

5 Jun 2023

Reponse to the academic editor

Question 1: Please ensure that your manuscript meets PLOS ONE's style requirements, including those for file naming.

Answer 1: I have check my manuscript according to PLOS ONE's style requirements,and revised some incorrect formatting.

Question 2: We noticed you have some minor occurrence of overlapping text with the following previous publication(s), which needs to be addressed.

Answer 2:when I check thesis with Paper YY, repetition rate was 8.0% with the two articles you provided. I have revised these similarities.

Question 3: We note that the grant information you provided in the ‘Funding Information’ and ‘Financial Disclosure’ sections do not match. When you resubmit, please ensure that you provide the correct grant numbers for the awards you received for your study in the ‘Funding Information’ section.

Answer 3: I am sorry for my ignore to provide incorrect Funding Information, When I resubmit, I have provided the correct grant numbers for the awards. 

Question 4: We note that you have stated that you will provide repository information for your data at acceptance. Should your manuscript be accepted for publication, we will hold it until you provide the relevant accession numbers or DOIs necessary to access your data. If you wish to make changes to your Data Availability statement, please describe these changes in your cover letter and we will update your Data Availability statement to reflect the information you provide.

Answer 4:Thank you for your good advice, I wouldn’t like to change my data availability statement.

Question 5: PLOS requires an ORCID iD for the corresponding author in Editorial Manager on papers submitted after December 6th, 2016. Please ensure that you have an ORCID iD and that it is validated in Editorial Manager. To do this, go to ‘Update my Information’ (in the upper left-hand corner of the main menu), and click on the Fetch/Validate link next to the ORCID field. This will take you to the ORCID site and allow you to create a new iD or authenticate a pre-existing iD in Editorial Manager. Please see the following video for instructions on linking an ORCID iD to your Editorial Manager account

Answer 5: Thank you for your good advice, I have got ORCID iD, as follows: https://orcid.org/0000-0002-0968-7270.

Question 6: We note that you have provided funding information that is not currently declared in your Funding Statement. However, funding information should not appear in the Acknowledgments section or other areas of your manuscript. We will only publish funding information present in the Funding Statement section of the online submission form.

Answer 6:I am sorry for my ignore for providing funding information in acknowledgments section, it has been removed , and acknowledgments section has been improved, as follows : 

“we would like to thank the Biomedical Technology Co from Shanghai Ouyi for special fungal technical support and remote guidance”.

Question 7: Please include your amended statements within your cover letter; we will change the online submission form on your behalf.

Answer 7:Thank your for good advice, amended funding statements as follows:

We gratefully acknowledge the ﬁnancial support of the National Natural Science Foundation of China (Grant No. 31560003) and the Young Scholars Science Foundation of Lanzhou Jiaotong University (No.2022020),which has been added in cover letter with red font.

Reponse to the reviewer 1

Question 1: In M&M, please indicate clearly how many biological replicates have been analysed, and how the biological replicates were obtained.

Answer 1: I am sorry for my ignore, I have added this point in M&M(Page 5,Line 89-91), as follows:

80 mg different treatment materials were taken and put them in liquid nitrogen , then placed in -80℃ refrigerator, immediately. Three biological duplicate samples from each treatment group.

Question 2: Fig 3: add statistical analysis.

Answer 2:Thank you for your good advice,it is necessary to add statistical analysis in Fig 3. I have added significant difference analysis in Fig 3. In addition, Statistical analysis labeling have been added in Fig 3 legand(Page 16,Line 312-313). as follows:

Error bars indicate the mean standard error calculated from three biological replicates. The statistical level was according to ∗ P < 0.05, ∗∗P < 0.01.

Question 3: Throughout the manuscript, please change qRT-PCR to RT-qPCR.

Answer 3: I am sorry formy ignore, I have checked the whole manuscript and changed qRT-PCR to RT-qPCR.

Question 4:In the abstract, indicate in full the names of the genes.

Answer 4:Thank you for your good advice, I am sorry for my omission. I have added full names of the genes in abstract section.

Reponse to the reviewer 2

Major issues:

Question 1:The expression of 16 candidate genes must be supplied from the transcriptome data in the present study. The sequence of 16 selected transcripts needs to be given as supplementary.

Answer 1:Thank you for your good advice, it is essential to supply sequence of 16 selected transcripts in our study, now it has been given as supplementary(Page 6, Line 117). 

Question 2:How much amount of RNA was taken for cDNA synthesis?

Answer 2: I am sorry for my ignore, The RT-qPCR mixture for cDNA synthesis contained 0.5 μL RNA, 5μL 5×TransScript All-in-One SuperMIX for qPCR; 0.5 μL gDNA Remover and 4 μL nuclease-free H2O.Which has been improved in method section(Page 5, Line 102-103).

Question 3:I assume there must be some published references for RNA extraction in this specie, as the genome is already available. Can you cite them in the material method section?

Answer 3:Thank you for your good advice, I also think it is necessary. I have cited published references in the material method section(Page 5, Line 97).

Question 4:2-delta delta Ct reference missing

Answer 4: I am sorry for my ignore, the reference about the calculation method of 2-∆∆CT has been supplemented(Page 6, Line 114).

Question 5:I recommend adding a table with 16 selected genes and their transcriptomic expression in the different conditions in Ophiocordyceps sinensis (as per availability) with a putative function and reference for the readers.

Answer 5:That’s a good advice to add a table with 16 selected genes and their transcriptomic expression in the different conditions in Ophiocordyceps sinensis (as table), I have done it. But sorry for my stupid, I am trouble in how to show putative function and reference for the readers. Would you give me some advice?

Gene Different Primordia Developmental Stages(FRKM) Diﬀerentially Morphologic Mycelium Stages(FRKM) Different Stresses(FRKM)

 mycelia primordia fruit bodies aeria hyphae knot mycelium 4℃ light(300 lx) NaCl (3.8 %)

ACT1 11.2 16.4 20.4 5.4 13.5 21.1 13.3 16.4 11.3

β-TUB 15.4 18.6 22.2 12.2 14.7 11.1 14.3 15.8 16.3

18S rRNA 20.6 23.3 25.6 17.7 29.5 21.3 18.8 20.2 19.4

RPL2 6.3 12.5 18.3 7.7 8.9 10.1 5.1 8.8 12.4

UBQ 10.3 11.6 10.8 8.5 9.4 8.1 8.8 9.6 6.7

EF1-α 13.2 14.1 13.7 10.3 11.8 9.1 12.2 14.3 16.9

PGM 5.4 9.5 13.1 2.3 3.1 2.8 6.6 9.8 5.1

TPI 3.56 12.6 20.2 5.5 21.3 15.8 5.1 8.4 12.2

QPRTase 4.4 19.5 7.8 7.6 18.6 30.3 6.1 11.8 16.2

H+-ATPase 7.54 12.98 17.09 5.4 12.6 19.8 9.8 18.2 14.4

PGI 10.9 20.4 4.3 6.5 8.9 4.4 8.3 16.6 21.2

GGT 4.5 14.5 25.1 9.5 13.7 18.1 12.2 17.3 21.3

TYR1 2.7 6.3 9.5 5.5 18.8 29.4 3.6 7.7 12.1

CDC14 15.5 21.3 34.4 10.3 21.1 27.4 17.8 25.6 33.1

CYS 27.4 36.54 17.43 15.6 19.8 25.1 20.3 29.8 30.3

GAPDH 4.3 9.6 7.5 3.3 18.6 32.1 8.8 19.4 15.4

Question 6:Line 123- 130 is unnecessary as the full name of the genes is given in the table1.

Answer 6: Thank you for your good advice, it does seem a bit redundant, and I've removed it from Table 1.

Question 7: The melting curve picture should be supplementary. Try to give an agarose gel picture of the PCR-amplified target genes (16) as a supplementary.

Answer 7:Thank you for your good advice, an agarose gel picture of the PCR-amplified target genes (16) has been given combined with the melting curve picture as S1 Fig. In addition, the related content has been suppled in results section (Page 7, Line 138-140). As follows:agarose gel electrophoresis results showed all of sixteen CRGs were specifcally amplifed a single product with specific size (S1 Fig 1B).the related Fig legand has been suppled in supporting information(Page 16, Line 320-323), as follows:

S1 Fig. Dissolution curves and amplification products and of sixteen CRGs. A . Dissolution curves and amplification products and of sixteen CRGs. B. Amplifcation products of 16 CRGs: DNA marker, DL2000; 1–16, 18S rRNA, QPRTase, β-TUB, RPL2, EF1-α, PGI, PGM, H+-ATPase, ACT1, UBQ, GAPDH, CYS, GGT, TPI, TYR1and CDC14.

Question 8:Line 243: either write other fungal species or if you want to make a general comment like “in organisms” then add some recent citations

Answer 8: Thank you for your good advice, adding some recent citations in which “organisms” was necessary. I have added some recent citations in manuscript (Page 12, Line 251). In addition, corresponding literatures have been updated in reference.

Question 9:There is some redundancy in different manuscript sections; try to minimize those. For example, line 277 and line 281 redundancy mentions the same thing.

Answer 9: Thank you for your good advice, I have check the whole manuscript,and revised some cumbersome place

Question 10:Mention the function of the Oscox gene somewhere in the manuscript to prepare the ground for the readers to digest the result and the rationality of it during the validation of the selected gene section.

Answer 10: Thank you for your good advice, it was necessary to add the ground of Oscox gene for the readers to digest the result and the rationality, Oscox means Cytochrome oxidase in O.sinensis, which has been applied in results (Page 12, Line224).

Minor issues

Question 1: English is very poor. I recommend professional linguistic correction before resubmission.

Answer 1: I am sorry for my poor english, thank your for your good advice, I have asked help for full text revision from my english professional friends .

Question 2: The short forms of gene names for less common genes are often hard to follow for non-field readers.

Answer 2:I am sorry for my ignore, I have supplied full name of all genes in manuscript.

Question 3:The species names should be in Italics throughout the text.

Answer 3:I am sorry for my ignore, The species names have been revised in Italics throughout the text.

Question 4:Do not use the short form in the abstract. Alternatively, short forms must appear with the full form in the first appearance in the text.

Answer 4:Thank you for your good advice, I have supplied the whole name of short form in the abstract.

Question 5:Line 43 and Line 47: qPCR or qRT-PCR?

Answer 5: I am sorry for the trouble due to my negligence, it should be RT-qPCR,which has been revised.

Question 6:Line 91: Xie`s study? Please refer to the article.

Answer 6:I am sorry for my ignore,the related reference has been supplied (Page 5, Line95)

Question 7:Word spacing needs to be corrected throughout the text. The authors are not very carefully edit the manuscript before submission. (i.e., line 26, 89,199).

Answer 7: I am sorry for my ignore, Word spacing have been corrected throughout the text.

Question 8:Line 198: correct (“and and”)

Answer 8: I am sorry for my inadvertent, it has been deleted.

Question 9:Line 206: validation section: try to mention the most stable or least stable gene in the bracket while explaining the result. It is interesting but hard to follow in its present form.

Answer 9: Thank you for your good advice,it is necessary to add the most stable gene in the bracket while explaining the result,which has been supplied in validation section (Page 13, Line231-233, Line243-244, Line255-256, )

---

## [Editor Report · Decision Letter 1]

8 Jun 2023

PONE-D-22-28076R1Selection and validation of reference genes for RT-qPCR in ophiocordyceps sinensis under diﬀerent experimental conditionsPLOS ONE

Dear Dr. Li,

Thank you for submitting your manuscript to PLOS ONE. After careful consideration, we feel that it has merit but does not fully meet PLOS ONE’s publication criteria as it currently stands. Therefore, we invite you to submit a revised version of the manuscript that addresses the points raised during the review process.

We look forward to receiving your revised manuscript.

Kind regards,

Xiaoping Pan

Academic Editor

PLOS ONE

Journal Requirements:

Additional Editor Comments:

This manuscript was revised well. However, I found some references were cited wrong. For example, RefFinder program was developed by Dr. Zhang's lab, please see the publications about this RefFinder program: Xie, F., Wang, J. & Zhang, B. RefFinder: a web-based tool for comprehensively analyzing and identifying reference genes. Funct Integr Genomics 23, 125 (2023). https://doi.org/10.1007/s10142-023-01055-7; and Xie F, Xiao P, Chen D, Xu L, Zhang B (2012) miRDeepFinder: a miRNA analysis tool for deep sequencing of plant small RNAs. Plant Mol Biol 80(1):75–84. https://doi.org/10.1007/s11103-012-9885-2. Please correct this references and give the credit to the right person. Please also carefully check other references.

---

## [Author Response · Author response to Decision Letter 1]

12 Jun 2023

Reponse to the journal requirements and Additional Editor Comments

Reponse to journal requirements

Question 1: Please review your reference list to ensure that it is complete and correct. If you have cited papers that have been retracted, please include the rationale for doing so in the manuscript text, or remove these references and replace them with relevant current references. Any changes to the reference list should be mentioned in the rebuttal letter that accompanies your revised manuscript. If you need to cite a retracted article, indicate the article’s retracted status in the References list and also include a citation and full reference for the retraction notice. 

Answer 1: Thank you for your good advice,I have check the reference list of my manuscript. All cited papers were complete and correct except for reference 20, due to this paper was a Chinese journal, so it cannot be identified,while this cited paper was important for the background construction.

Reponse to Additional Editor Comments:

Question 1: This manuscript was revised well. However, I found some references were cited wrong. For example, RefFinder program was developed by Dr. Zhang's lab, please see the publications about this RefFinder program: Xie, F., Wang, J. & Zhang, B. RefFinder: a web-based tool for comprehensively analyzing and identifying reference genes. Funct Integr Genomics 23, 125 (2023). https://doi.org/10.1007/s10142-023-01055-7; and Xie F, Xiao P, Chen D, Xu L, Zhang B (2012) miRDeepFinder: a miRNA analysis tool for deep sequencing of plant small RNAs. Plant Mol Biol 80(1):75–84.https://doi.org/10.1007/s11103-012-9885-2.Please correct this references and give the credit to the right person. Please also carefully check other references.

Answer 1: I am sorry for my ignore. Firstly, I have replaced references 25 with your advised (https://doi.org/10.1007/s10142-023-01055-7).Then, I have also carefully check the other references, I found reference 10 was unsuitable and it has been replaced.

Question 2:While revising your submission, please upload your figure files to the Preflight Analysis and Conversion Engine (PACE) digital diagnostic tool, https://pacev2.apexcovantage.com/. PACE helps ensure that figures meet PLOS requirements

Answer 2: The image format detection of Fig 1, Fig 2 and S1 Fig have been carried out with he Preflight Analysis and Conversion Engine (PACE) digital diagnostic tool, among which,Fig 1 reached figure file requirements, Fig 2 and S1Fig have been revised and upload again.

---

## [Editor Report · Decision Letter 2]

16 Jun 2023

Selection and validation of reference genes for RT-qPCR in ophiocordyceps sinensis under diﬀerent experimental conditions

PONE-D-22-28076R2

Dear Dr. Li,

We’re pleased to inform you that your manuscript has been judged scientifically suitable for publication and will be formally accepted for publication once it meets all outstanding technical requirements.

Kind regards,

Xiaoping Pan

Academic Editor

PLOS ONE
---

## [Editor Report · Acceptance letter]

17 Aug 2023

PONE-D-22-28076R2 

Selection and validation of reference genes for RT-qPCR in *ophiocordyceps sinensis* under diﬀerent experimental conditions 

Dear Dr. He:

I'm pleased to inform you that your manuscript has been deemed suitable for publication in PLOS ONE. Congratulations! Your manuscript is now with our production department. 

Kind regards, 

on behalf of

Dr. Xiaoping Pan 

Academic Editor

PLOS ONE